# The Impact of COVID-19 Vaccination on Oxidative Stress and Cardiac Fibrosis Biomarkers in Patients with Acute Myocardial Infarction (STEMI), a Single-Center Experience Analysis

**DOI:** 10.3390/life14111350

**Published:** 2024-10-22

**Authors:** Razan Al Namat, Letiția Doina Duceac, Liliana Chelaru, Cristina Dimitriu, Amin Bazyani, Andrei Tarus, Alberto Bacusca, Adrian Roșca, Dina Al Namat, Lucian Ionuț Livanu, Elena Țarcă, Grigore Tinică

**Affiliations:** 1Faculty of Medicine, “Grigore T. Popa” University of Medicine and Pharmacy, 700115 Iassy, Romania; al-namat.razan@umfiasi.ro (R.A.N.); liliana.chelaru@umfiasi.ro (L.C.); c_dimit@yahoo.com (C.D.); aminbazyani@gmail.com (A.B.); andrei.tarus@umfiasi.ro (A.T.); alberto-emanuel-l-bacusca@d.umfiasi.ro (A.B.); grigore.tinica@umfiasi.ro (G.T.); 2Faculty of Medicine and Pharmacy, “Dunărea de Jos” University, 800008 Galati, Romania; letimedr@yahoo.com; 3“Saint Mary” Emergency Children Hospital, 700309 Iassy, Romania; adrianrosca82@yahoo.com; 4Department of Surgery II—Pediatric Surgery, “Grigore T. Popa” University of Medicine and Pharmacy, 700115 Iassy, Romania; namat@d.umfiasi.ro; 5Faculty of Medicine and Pharmacy, “George Emil Palade” University of Medicine, Pharmacy, Science and Technology of Târgu Mureș, 540139 Târgu Mureș, Romania; ionutlivanu97@gmail.com

**Keywords:** acute myocardial infarction (STEMI), oxidative stress, cardiac fibrosis, COVID-19 vaccination, SARS-CoV-2 infection

## Abstract

The relationship between the classical cardiac biomarker and acute myocardial infarction (STEMI) in patients with COVID-19 is far from being elucidated. Furthermore, superoxide dismutase (SOD), a marker for oxidative stress, was associated with cardiac ischemia. Also, Galectin-3 is significant for defining the relationship between cardiac fibrosis and COVID-19. There are no studies on the effect of SARS-CoV-2 virus infection and vaccination on patients with STEMI and biomarkers above-mentioned. Aim: our single-center prospective study assesses the relationship between COVID-19 infection with/without vaccination and the value of SOD and Galectin-3 in STEMI patients. Material and methods: In total, 93 patients with STEMI and SARS-CoV-2 virus infection were included in the analysis, patients were divided in two groups based on COVID-19 vaccination status. Echocardiographic and laboratory investigations for cardiac ischemia, oxidative stress, and cardiac fibrosis biomarkers were investigated. Results: In total, 93 patients were included, the majority of which were male (72.0%), 45.2% (*n* = 42) were vaccinated against SARS-CoV-2; the mean age of vaccinated patients is 62 years, and 57% (*n* = 53) are smokers; blood pressure is found with a higher frequency in unvaccinated people (62.7%) compared to 28.6% in vaccinated people (*p* = 0.015), and 90.5% of the vaccinated people presented STEMI, compared with 96.1% of the unvaccinated ones. Revascularization with one stent was achieved in 47.6% of the vaccinated people and 72.5% for the unvaccinated people (*p* = 0.015). Galectin-3 was slightly more reduced in the vaccinated patients compared to the unvaccinated patients (0.73 vs. 0.99; *p* = 0.202), and the average level of Cu/ZnSOD was slightly more reduced in vaccinated patients compared to the unvaccinated patients (0.84 vs. 0.91; *p* = 0.740). Conclusions: Regarding patient’s SARS-CoV-2 infection functional status, the results from our single-center analysis did not find a statistically significant decrease in oxidative stress and cardiac fibrosis biomarkers along with cardiovascular complication following STEMI treated with percutaneous coronary angioplasty (PCI) in the case of patients with COVID-19 vaccination compared with patients who did not receive COVID-19 vaccine. Anyway, our data suggest that contemporary PCI techniques may offer an alternative revascularization strategy that enables complex CAD COVID-19 patients to be safely discharged from hospital.

## 1. Introduction

### 1.1. General Considerations on the SARS-CoV-2 Vaccination and Its Impact on the Cardiovascular System

January 2020 was the start for the global development of vaccines against the genome sequencing of the coronavirus 2 (SARS-CoV-2) severe acute respiratory syndrome, the etiologic agent of coronavirus disease 2019 (COVID-19). Significant concern was raised with regard to the safety of the new coronavirus vaccines, mostly because of the rapid vaccine research and development process [1]. A survey carried out in August 2020 showed that 70% of the participants were concerned about the immediate development of vaccines. Then, a review conducted in September 2021 emphasized the safety concerns predicting hesitance [2]. Next, rare adverse events post-immunization were reported further to the extensive vaccination programs [3]. The highest number of cardiovascular (CV) adverse events in patients receiving the BNT162b2 (Pfizer-BioNTech) COVID-19 vaccine was shown in a survey based on the World Health Organization (WHO) database (VigiBase). In total, 30% and 44% of CV adverse events were described as severe following vaccination by BNT162b2 (Pfizer-BioNTech) and mRNA-1273 (Moderna). Both vaccines caused palpitations and tachycardia, as common CV adverse events [4,5,6]. In addition, a large group of participants without any prior cardiac disease suffered from myocarditis (*n* =  15) following mRNA COVID-19 vaccination [7]. Later, a time relationship of myocarditis in young males following Pfizer-BioNTech vaccination appeared in another large-scale study in the healthcare field [8].

There are also studies showing the potential mechanism of post-COVID-19 vaccine CV adverse events, though the CV pathogenesis remains unclear. It is possible that the mRNA vaccine triggers an immune response where mRNA is detected as an antigen in genetically predisposed individuals [9]. When inflammatory cascades are activated, following the expression of cytokines by dendritic and Toll-like receptors, an immunomodulatory response against the mRNA occurs, and this potentially results in myocarditis and other systemic reactions [10]. Some surveys showed that macrophage activity was increased by post-vaccination immune thrombocytopenic purpura (ITP) and platelet production in patients with mild “compensated” thrombocytopenia or chronic, hereditary thrombocytopenia was reduced [11]. Postmortem findings of patients with vaccine-induced thrombotic thrombocytopenia (VITT) reflected that the ITP after vaccination consisted in thrombosis at uncommon sites, the splanchnic, adrenal, cerebral, and ophthalmic veins [12,13,14,15]. 

At the end of April 2021, the Israeli Ministry of Health reported myocarditis in people vaccinated with BNT162b2 (Pfizer-BioNTech). Later, the Vaccine Adverse Event Report System (VAERS) database of the FDA (Food and Drug Administration) was checked to revise the side effects following COVID-19 immunization; 45 cases of myocarditis were identified, 19 deriving from BNT162b2 (Pfizer-BioNTech) and 26 from mRNA-1273 (Moderna) [16]. As a result, an investigation was initiated by the Centers for Disease Control and Prevention (CDC) upon the patients with reported myocarditis after vaccination, with a view to identifying the long-term impact of the cardiac complications caused by the two approved COVID-19 mRNA vaccines [17]. 

### 1.2. Study Aims

This study contributes to the growing scientific literature on the COVID-19 pandemic by assessing the impact of COVID-19 vaccination on systemic inflammation, oxidative stress (superoxide dismutase- SOD), and myocardial fibrosis markers (Galectin-3) in patients with acute myocardial infarction (STEMI).

## 2. Materials and Methods

### 2.1. Overview of Study Coordinates

The study enrolled consecutive consenting patients vaccinated and unvaccinated against COVID-19 infection, admitted for acute myocardial infarction (STEMI), with past or present confirmation of COVID-19 infection at the “Prof. Dr. George I.M. Georgescu” Institute of Cardiovascular Diseases in the NE Romanian city of Iași from 2021 to 2023. Formal approval for the study was obtained from the Ethics Committee of the “Grigore T. Popa” University of Medicine and Pharmacy Iași, and the general provisions of the Helsinki Declaration regarding medical research on human subjects were observed.

The study was conducted prospectively over a period of 24 months. In addition, medical history data were retrieved from the patients’ records. In the first hours following the intervention, samples were collected for routine tests and for the purpose of measuring oxidative stress SOD and Galectin-3 levels. 

### 2.2. Inclusion and Exclusion Criteria

Patient were enrolled based on the following criteria: written informed consent; admitted with STEMI in the Clinic of Cardiovascular Surgery of the Institute of Cardiovascular Diseases of Iasi, Romania; post-COVID-19 status about 3 months prior to this major acute cardiovascular event (MACE); lipid and glucose metabolism disorders; body mass index (BMI) > 25 kg/m^2^; aged 40–90 years old. Exclusion criteria were as follows: patients admitted in the Clinic of Cardiovascular Surgery of the Institute of Cardiovascular Diseases of Iasi for other conditions than myocardial infarction; no prior COVID-19 infection; psychiatric conditions; lack of compliance to any item of the study demands; cancer disease in evolution.

### 2.3. Patient Data Collected and Definitions

The collected data were organized into sets and series as it can be seen in the Results section for the studied variables; data regarding general patient characteristics and past medical histories, clinical, paraclinical, echocardiographic, and laboratory findings, oxidative stress (SOD), and the fibrosis biomarker Galectin-3 was collected. The serum markers of fibrosis (Galectin-3) was assessed by the ELISA technique after venous blood collection in vacutainer tubes without anticoagulant and centrifugation and the serum markers of oxidative stress were assessed by spectrophotometry. Anti-SARS-CoV-2 antibodies (IgG) were measured by qualitative detection in human serum by the ELISA method. All blood venous samples were transported within 2 h of collection and processed after storage in refrigerator (−200 °C).

### 2.4. Organization of Database and Statistical Analysis

The patient data were first anonymized and collected in Microsoft Office Excel version 2010, then processed in IBM SPSS Statistics for Windows, version 20 (IBM Corp., Armonk, NY, USA). Data series were organized in the following two main study groups (STEMI patients with/without previous COVID-19 vaccination) and six subgroups based on the severity of post-COVID-19 functional status.

STEMI patients with previous COVID-19 vaccination:
a.With mild post-COVID-19 functional status;b.With moderate post-COVID-19 functional status;c.With severe post-COVID-19 functional status.STEMI patients without previous vaccination anti COVID-19:
a.With mild post-COVID-19 functional status;b.With moderate post-COVID-19 functional status;c.With severe post-COVID-19 functional status.

## 3. Results

### 3.1. General Patient Characteristics in Relation with COVID-19 Vaccination

In the table below, we present demographical data and comorbidities (Table 1 and Table 2). In total, 93 patients were included, the majority of which are male (72.0%); the median age of vaccinated patients is 62 years old, while the median age of the unvaccinated population is 57 years old. You can notice that 57% (*n* = 54) of the studied group are smoking patients, who are, in their turn, distributed in a percentage of 45.2% (*n* = 19) as being vaccinated, and 66.7% (*n* = 34) unvaccinated. Regarding a few of the comorbidities selected in the study, we notice that stage 1 hypertension, which is a risk factor for myocardial infarction, occurs in a higher frequency in unvaccinated persons (62.7%) compared to 28.6% in the vaccinated ones (*p* = 0.001). This is also valid for stage 2 hypertension, which is more common in vaccinated people (40.5%) compared to the unvaccinated ones (13.7%) (*p* = 0.004). If we look at CKD, from a total of 13 patients, the vaccinated ones, no matter the kidney disease degree, have associated more frequently CAD that required aorto-coronary bypass (*n* = 10), compared to the unvaccinated ones with CKD (*n* = 3), with a statistical significance (*p* = 0.012)—see Table 1.

Heart failure within vaccinated and unvaccinated patients was distributed in the 4 NYHA severity forms. We can notice that within the vaccinated group, most patients were associated with NYHA II heart failure (42.9%), followed by NYHA III heart failure (38.1%). Also, a small percentage presented the worst form of heart failure (7.1%). Regarding the group of unvaccinated patients, the hierarchy of heart failure severity was similar: NYHA II (49%); NYHA III (23.5%); NYHA IV (5.9%).

Another comorbidity included in the study was diabetes mellitus. Only a total of seven (7.5%) people of the studied group presented diabetes mellitus as a diagnosis: 7.1% of the vaccinated ones and 7.8% of the unvaccinated ones; however, without having any statistical relevance (*p* = 0.990). However, a more significant number of people presented low tolerance to glucose: 42.9% of the total vaccinated people, compared to 43.1% of the unvaccinated ones (*p* = 0.898).

From a pulmonary point of view, 22.3% of the total studied group presented a history of pulmonary diseases, others than the COVID-19 infection. Their distribution was approximately similar within the vaccinated subgroups (23.8%) and unvaccinated subgroups (23.5%) (*p* = 0.75). Regarding COVID-19 infection history, with a statistically significant value (*p* = 0.001), it occurred in a total of 62.4% of people from the entire study group. Among the vaccinated ones, 95.2% suffered from a COVID-19 form prior to hospital admission, while among the ones who chose to not receive vaccination, this number was only 35.3%. 

Among the 93 patients introduced in the study, a percentage of 45.2% (*n* = 42) were vaccinated against SARS-CoV-2, and 54% (*n* = 51) are represented by those who refused vaccination. Of the 45.2% of patients who accepted to receive vaccination, 11.8% received only one dose, 23.7% were vaccinated with two doses, while 9.7% were vaccinated with three doses—see Figure 1.

Of the 42 vaccinated patients, 10.8% (*n* = 10) chose to receive vaccination with Astrazeneca, 12.9% (*n* = 12) chose to receive vaccination with Johnson, 2.2% (*n* = 2) chose to receive vaccination with Moderna, while most of them, 19.4% (*n* = 18), chose to receive vaccination with Pfizer. In total, 54.8% refused to receive vaccinatiom—see Figure 2.

Regarding symptomatology, cough and dyspnea were the variables studied. In both cases, the vaccinated group presented a higher frequency for the two symptoms, compared to the unvaccinated group, respectively, 31% and 33.3% compared to 17.6%; however, without any statistical relevance. In terms of complications, the most notable difference from a statistical point of view, with *p* = 0.010, is represented by the hospitalization days. As a result, the vaccinated patients had an average of 7.12 ± 2.57 compared to the unvaccinated group, with an average of 7.12 ± 2.57 compared to the unvaccinated group with an average of 6.35 ± 3.42 days—see Table 2.

Among complications, the depression and anxiety scores were assessed with the aid of the Hamilton scale. Thus, we could conclude that in the group of vaccinated patients, the HAM-D average was 13.33 ± 3.06 points, and within the unvaccinated group, the average was 14.33 ± 3.52 points, which meant a slight depression with *p* = 0.152. Also, with respect to the anxiety level, both subgroups associated scores that signify the slightly moderate form of anxiety (HAM-A = 21.9 ± 3.36 points—vaccinated; HAM-A = 21.98 ± 3.36 points—unvaccinated).

Regarding the type of surgical intervention, 2.4% of the vaccinated patients have undergone a coronary bypass compared to the 5.9% of the unvaccinated ones (*p* = 0.394). The interventions on valvulopathies were performed in a percentage of 2.0%, only within the unvaccinated people (*p* = 0.217).

### 3.2. Echocardiographic and Laboratory Findings 

From the point of view of echocardiographic parameters, we noticed a significantly higher telediastolic diameter of the left ventricle in the vaccinated group (51.79 ± 5.06 mm) compared to the unvaccinated group (47.24 ± 12.87 mm, *p* = 0.034). The heart pumping function was assessed by echocardiographic examination, emphasizing a slightly better ejection fraction within the vaccinated patients’ group (EF 42.07% ± 10.46%). The group of unvaccinated patients had an average ejection fraction of 40.39% ± 11.60%—see Table 3.

The distribution regarding the interventional treatment of both vaccinated and unvaccinated patients is presented in Table 4. We can notice that 91.4% of the patients benefited from interventional treatment. In total, 90.5% of the vaccinated people presented with STEMI, compared to 96.1% of the unvaccinated group. The non-STEMI number is higher, with a distribution of 9.5% within the vaccinated group and 3.9% within the unvaccinated group. The revascularization with one stent was successfully achieved in 47.6% of the vaccinated patient group, while within the unvaccinated patient group, the percentage was higher, with 72.5% (*p* = 0.015). On the other hand, the vaccinated patients and those requiring two stents were in a percentage of 33.3%, compared to 7.8% of the unvaccinated patients (*p* = 0.002)—see Table 4.

Concerning the biological profile, we studied an important set of parameters described in Table 5, some of which are described in more detail below. K level was within normal parameters in both groups, vaccinated or unvaccinated (*p* = 0.047). In addition, the Mean Corpuscular Volume, the parameter used in classifying the anemias, had an average value of 120.42 f ± 23.28 in the vaccinated group, and in the unvaccinated group the average value was 87.83 ± 5.27 (*p* = 0.046).

Another studied parameter, but without statistical correlation, was the activity of superoxide dismutase—analysis that quantifies the level of oxidative stress. It had a value of 0.84 ± 1.02 ng/mL in the vaccinated subgroup and 0.91 ± 1.13 ng/mL in the unvaccinated subgroup. Galactin-3 is a ß-galactosidase binding lectin, important in numerous biological activities of different organs, among which is the early occurrence of fibrosis. Its levels within vaccinated patients (0.73 ± 0.20 ng/mL) were lower, compared to the unvaccinated group (0.99 ± 0.18 ng/mL); however, there was no statistical significance—see Table 5.

The hepatic function was analyzed in the two subgroups, by dosing GPT, GOT, and GGT, without being significantly influenced from a statistical point of view by the vaccination status (*p* = 0.084; *p* = 0.406; *p* = 0.128). Also, for the inflammatory status, we dosed CRP and Fibrinogen, with an average value of 27.02 ± 6.99 mg/dL and 493.05 ± 149.54 mg/dL within the vaccinated patients, respectively, and a value of 31.73 ± 5.79 mg/dL and 499.43 ± 124.53 mg/dL in unvaccinated patients, respectively, without having a statistical relevance (*p* = 0.602; *p* = 823).

Regarding the lipid profile, we analyzed the biological parameters, such as total Cholesterol together with its fractions HDL-c and LDL-c, and the triglyceride levels as well. All their results, in both studied subgroups, vaccinated and unvaccinated, were approximately similar without a statistical significance (*p* > 0.05).

The blood coagulation function, measured by INR dosing, was not influenced by the relationship with the COVID-19 vaccination (*p* = 0.740). The renal function was within normal parameters in both groups, as well, without being influenced by the vaccine (*p* = 0.71).

The average level of Galectin-3 was slightly more reduced in the vaccinated patients compared to the unvaccinated patients (0.73 vs. 0.99; *p* = 0.202), and the average level of Cu/ZnSOD was slightly more reduced in vaccinated patients compared to the unvaccinated patients (0.84 vs. 0.91; *p* = 0.740)—see Figure 3.

SYNTAX score was 27.58 ± 2.87 in vaccinated patients vs. 26.76 ± 2.68 in unvaccinated patients (*p* = 0.153)—see Figure 4 and Figure 5.

## 4. Discussion

In our study group, we included 93 patients, most of which were male (72.0%), and the median age of vaccinated patients is 62 years old, while the median age of the unvaccinated is 57 years old. We could conclude that 57% (*n* = 53) of the studied group are smokers, the latter being distributed in their turn into a percentage of 45.2% (*n* = 19) as vaccinated and 66.7% (*n* = 34) as unvaccinated. Regarding a few of the comorbidities selected in the study, we notice that Stage 1 Hypertension, which is also a risk factor for MI, is encountered with a higher frequency in unvaccinated people (62.7%) compared to 28.6% in the vaccinated people (*p* = 0.001); at the same time, if we look at CKD of the vaccinated people, no matter the severity of the kidney disease, they associated CAD with requiring aorto-coronary bypass (*n* = 10) more frequently, compared to the unvaccinated people with CKD (*n* = 3), with a statistical significance (*p* = 0.012) [18].

Since the onset of the SARS-CoV-2 infection, many medical, economic, social, and educational aspects have changed irreversibly [19]; from a medical point of view, numerous studies have been carried out and many articles have been published regarding the consequences of this epidemic on the health of the population. Researchers also published several systematic reviews on the relation between cardiovascular (CV) conditions and COVID-19 consequences. The most frequently occurring complications were cardiac complications, and cardiovascular disease is the most frequent comorbidity. Recent studies have shown that major complication after COVID-19 vaccination was the ischemic stroke, based on the number of outcome events during the 21-day risk interval [20]. Among individuals with cerebrovascular disorders (*n* = 286) and who had recently received mRNA-1273 (Moderna) or BNT162b2 (Pfizer-BioNTech) (*n* = 1,398,074), there were 246 cases of acute ischemic stroke. The fourth most common event in our study, myocarditis, has been increasingly reported in studies assessing the safety of mRNA-1273 (Moderna) and BNT162b2 (Pfizer-BioNTech), and the majority of cases occur after the second dose [21]. COVID-19 vaccine-associated myocarditis is usually temporary and self-limiting. Klein et al. reports postvaccination cases of myocarditis/pericarditis, of which 82% required hospitalization at a median length of 1 day. Two to three cases per million doses of myocarditis/pericarditis as adverse drug reaction in RNA-based vaccines have been found by a meta-analysis evaluating real-world data from the VAERS, managed by the CDC and FDA [22]. The European Agency of Medicines has officially reported that myocarditis is a side effect post-RNA vaccination, especially found among males, corresponding to 1.60 cases/million doses for Pfizer-BioNTech and 3.04 cases/million doses for Moderna in the region [23].

In the current study, heart failure within both the vaccinated and unvaccinated patients was distributed in the 4 NYHA severity degrees. We can notice that within the vaccinated group, most of them associated NYHA II heart failure (42.9%), followed by NYHA III heart failure (38.1%), and only a small part has experienced the most severe form of heart failure (7.1%). As to the group of unvaccinated patients, the hierarchy of heart failure severity hierarchy was similar: NYHA II (49%), NYHA III (23.5%), NYHA IV (5.9%), which confirms the data presented in recently published studies. Another comorbidity included in the study was diabetes mellitus. Only a total of seven people (7.5%) of the studied group presented diabetes mellitus as diagnosis, this was 7.1% of the vaccinated group and 7.8% of the unvaccinated group; however, without having any statistical relevance (*p* = 0.990). A more important number of people presented, in exchange, with a low tolerance to glucose: 42.9% of the total vaccinated people, compared to 43.1% of the unvaccinated people (*p* = 0.898) [24]. From a pulmonary point of view, 22.3% of the total studied group presented history of pulmonary diseases unrelated to COVID-19 infection. Their distribution was approximately similar within the vaccinated subgroups (23.8%) and unvaccinated subgroups (23.5%) (*p* = 0.75).

With respect to the history of COVID-19 infection, with a statistically significant value (*p* = 0.001), it existed in a total of 62.4% of people from the entire study group. Of the vaccinated patients, 95.2% had a form of COVID-19 prior to the hospital admission, while among those who chose to not become vaccinated, only 35.3% had a form of COVID-19 prior to the hospital admission. In terms of complications, the most significant difference from a statistical point of view, at *p* = 0.010, is represented by hospitalization days. As a result, the vaccinated patients had an average of 7.12 ± 2.57 compared to the unvaccinated group, with an average of 6.35 ± 3.42 days [25].

In relation to complications, we assessed the depression and anxiety score with the help of the Hamilton scale. Thus, we could conclude that in the group of vaccinated patients, the HAM-D average was of 13.33 ± 3.06 points, and within the unvaccinated patients the average was of 14.33 ± 3.52 points, which represents a slight depression with a *p* = 0.152. In addition, concerning the anxiety level, both subgroups associated scores that are classified into the slightly moderate form of anxiety (HAM-A = 21.9 ± 3.36 points—vaccinated; HAM-A = 21.98 ± 3.36 points—unvaccinated) [25,26].

Some underlying mechanisms have been proposed for acute myocardial infarction (AMI) following COVID-19 vaccination, with Kaposi Syndrome (KS) the most likely explanation—the coexisting occurrence of acute coronary syndromes with allergic reactions. The disease has been found in four variants: (1) type I: coronary spasm in patients with (nearly) normal coronary arteries; (2) type II: coronary thrombosis in patients with primary asymptomatic CHD; (3) type III: stent-related allergic coronary events, with IIIa (stent thrombosis) and IIIb (in-stent restenosis); and (4) type IV: anaphylaxis-mediated AMI in the patients with coronary grafts [27]. Aside from this, another possibility for AMI post-COVID-19 vaccination could be vaccine-induced thrombotic thrombocytopenia (VITT) associated thrombosis, although VITT might not develop in ten minutes only; therefore, this potential is very uncommon [28].

The following four types of mechanisms could help obtaining the pathophysiology of vaccine-induced allergic reactions. (1) Reactions through the pathway of mast cell activation and degranulation as IgE/antigen through cross-linking of FcεRI on mast cells [29]. (2) Another pathway that is performed via activation of the complement system is non-IgE-mediated mast cell degranulation, leading to the generation of anaphylatoxins C1q, C3a, C4, and C5a. (3) The directly activated Mas-related G protein-coupled receptor X2 (MRGPRX2) may mediate major allergic reactions and even mast cells via non-Fcε receptors, where the specific IgEs may not be detected, and the tryptase levels may show no abnormality in serious KS. Baronti et al. reported that the negative tryptase testing does not exclude allergic reactions. (4) Usually, the reaction starts 48 h post vaccination and reaches its peak between 72 and 96 h; it is cell-mediated and antibody independent, resulting from overstimulation of T cells and monocytes/macrophages, and cytokine releases causing inflammation, death of cells, and tissue damage [13,14,30].

Regarding the type of surgical intervention, 2.4% of the vaccinated patients underwent a coronary bypass compared to 5.9% of the unvaccinated ones (*p* = 0.394), although, SYNTAX score was 27.58 ± 2.87 in vaccinated patients vs. 26.76 ± 2.68 in unvaccinated patients (*p* = 0.153). The interventions on valvulopathies were conducted in a percentage of 2.0%, only within the unvaccinated patients (*p* = 0.271). We can notice that 91.4% of the patients benefitted from interventional treatment. In total, 90.5% of the vaccinated people presented with STEMI, compared to the 96.1% of the unvaccinated ones. Revascularization with one stent was successfully achieved within 47.6% of the group of vaccinated patients, while within the unvaccinated patients the percentage was higher at 72.5% (*p* = 0.015). On the other hand, the vaccinated ones and those who needed two stents were in a percentage of 33.3%, compared to 7.8% of the unvaccinated people (*p* = 0.002), which supports the data from the specialty literature [31].

While the significant demographical and clinical predictive factors are: female sex, older age, cigarette smoking, pre-existing medical conditions, absence of COVID-19 vaccination, infection with previous SARS-CoV-2 variants (i.e., pre-Omicron), acute symptoms, viral load, severe/critical COVID-19 illness, and invasive mechanical ventilation. There is more and more information that risk assessment may be sustained by the measurement of a few selected biomarkers, such as C-reactive protein and other inflammatory cytokines, lymphocyte count, lactate dehydrogenase, interferon γ, tumor necrosis factor α, and even fibrosis biomarkers, such as soluble suppression of tumorigenicity 2 and Krebs von den Lungen 6 [25,27,32]. There is a recent proteomic study, which emphasized that an important perturbation of the plasma proteome on the differential expression of proteins with involvement in lipid metabolism, complement and coagulation cascades, atherosclerosis, autophagy, and the lysosomal function, could foresee the persistence of COVID-19 symptoms up to 12 weeks following recovery with an accuracy of 94% [33].

Among other studied parameters that do not have a statistical correlation, there was the activity of superoxide dismutase, the analysis that quantifies the level of oxidative stress. This had a value of 0.84 ± 1.02 ng/mL in the vaccinated subgroup and 0.91 ± 1.13 ng/mL in the unvaccinated subgroup. Galactin-3 is a ß-galactosidase binding lectin, important in numerous biological activities of different organs, among which we mention the early occurrence of fibrosis. Its levels within vaccinated patients (0.73 ± 0.20 ng/mL) were lower compared to the unvaccinated ones (0.99 ± 0.18 ng/mL) without having a statistical significance. Also, for the inflammatory status, we dosed CRP and Fibrinogen, with a mean value of 27.02 ± 6.99 mg/dL and 493.05 ± 149.54 mg/dL within the vaccinated patients, respectively, and a value of 31.73 ± 5.79 mg/dL and 499.43 ± 124.53 mg/dLin the unvaccinated group, respectively; however, without any statistical relevance (*p* = 0.602; *p* = 0.823) [33,34].

The ReVR was a UK multicenter prospective registry that investigated the short-term outcomes of a novel cohort of patients with “surgical” CAD who would under normal circumstances be treated with CABG, but instead underwent PCI. When compared to historical PCI and isolated CABG reference groups, no significant differences in outcomes to hospital discharge were demonstrated other than a reduction in BARC 3–5 bleeding versus the CABG cohort [35,36].

Our data suggest that contemporary PCI techniques offer an alternative revascularization strategy that enables complex CAD COVID-19 patients to be safely discharged from hospital.

Ultimately, this syndrome cannot be considered a single clinical entity in terms of the care for patients with COVID-19 vaccination. It should be the topic for a more focused discussion and a cohesive multidisciplinary management customized to the type and severity of symptoms in correlation with comorbidities and type of vaccine. Rehabilitation has seemed to be the most effective treatment, but the current research will lead to the numerous multifaceted pathogenic mechanisms supporting the persistence of symptoms over a long time and eventually to a therapeutic strategy that could be personalized to individual care needs.

Study limitations include the small number of patients included and the small number of patients coming at cardiovascular control during COVID-19 pandemic. Potential study risks can be related to the completion of the studied group, the deterioration of blood, and the timely procurement of purchase kits and supplies; we are looking to continue our investigations in the near future including a large number of both groups or through conducting a multicenter study.

## 5. Conclusions

As literature reveals, the relation between the classical cardiac biomarker and acute myocardial infarction (STEMI) in patients with COVID-19 is far from being clarified. In addition, superoxide dismutase (SOD), a marker for oxidative stress, was associated with cardiac ischemia. In addition, Galectin-3 is highly important for defining the relation between cardiac fibrosis and COVID-19 [26,27,28]. The results from our single-center analysis show that there is no statistically significant decrease in oxidative stress and cardiac fibrosis biomarkers along with cardiovascular complication following STEMI treated with percutaneous coronary angioplasty (PCI) for patients with the COVID-19 vaccine versus the patients who did not receive the COVID-19 vaccine and especially for those candidates for CABG (high SYNTAX score >22 has the indication for CABG).

## Figures and Tables

**Figure 1 life-14-01350-f001:**
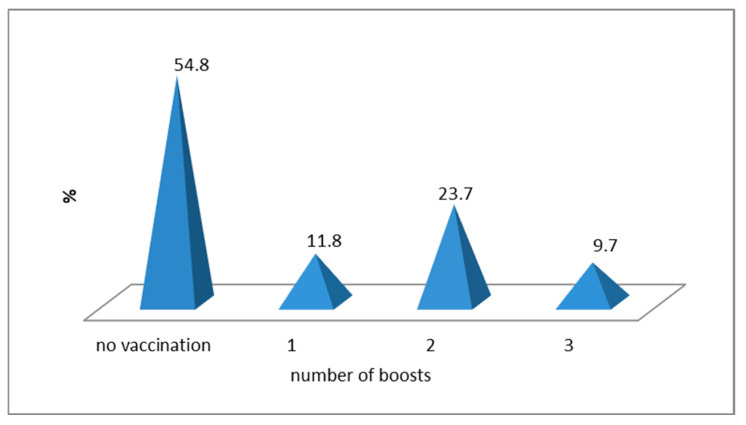
Distribution of patients according to anti-COVID-19 vaccination and number of doses.

**Figure 2 life-14-01350-f002:**
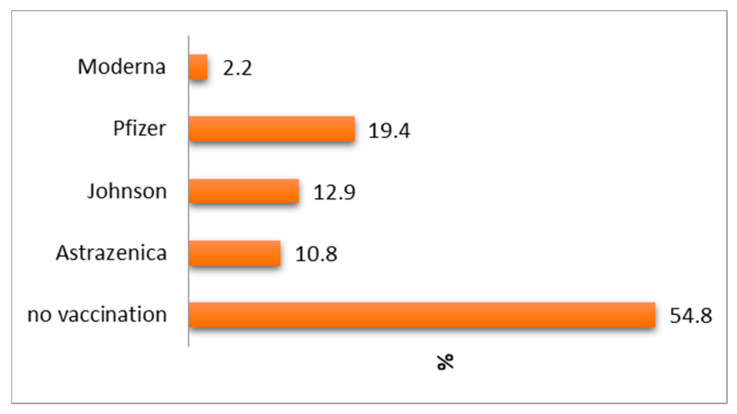
Distribution of patients according to the anti-COVID-19 vaccine type.

**Figure 3 life-14-01350-f003:**
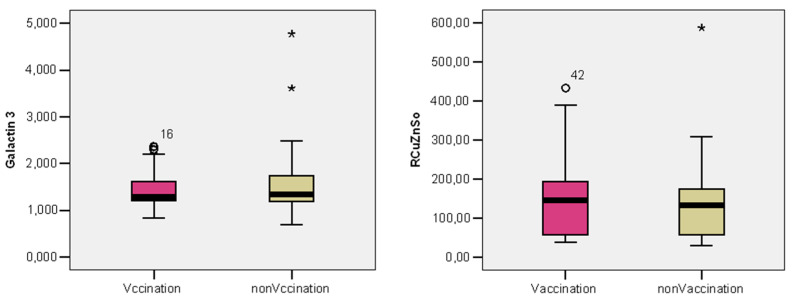
The distribution of patients according to Galectin-3 value and SOD in the vaccinated and unvaccinated groups.

**Figure 4 life-14-01350-f004:**
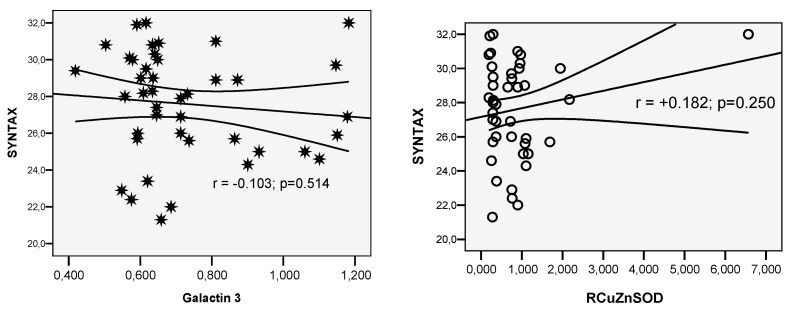
Correlation SYNTAX score with Galectin-3 value and SOD in the vaccinated group.

**Figure 5 life-14-01350-f005:**
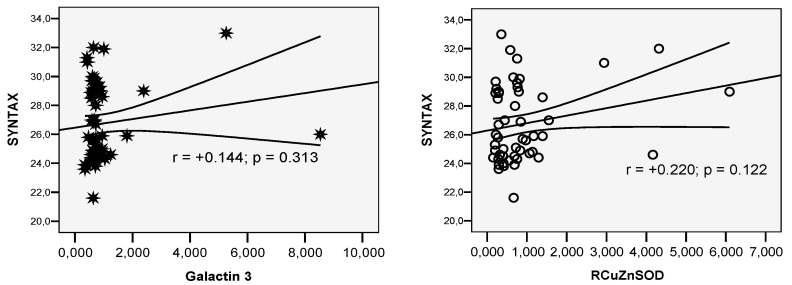
Correlation SYNTAX score with Galectin-3 value and SOD in the unvaccinated group.

**Table 1 life-14-01350-t001:** Baseline characteristics in relation with COVID-19 vaccination.

Characteristics	Total Cohort (*n* = 93)	Vaccination (*n* = 42)	Non-Vaccination (*n* = 51)	*p* Value for Chi-Square Test
**Demographic**				
Median age, y	59	62	57	0.021 ^(t)^
Male, *n* (%)	67 (72.0%)	29 (69.0%)	38 (74.5%)	0.560
Smoking, *n* (%)	53 (57.0%)	19 (45.2%)	34 (66.7%)	0.037
**Comorbidities**				
Obesity, *n* (%)	60 (64.5%)	27 (64.3%)	33 (64.7%)	0.966
HTN, *n* (%)	84 (90.3%)	38 (90.5%)	46 (90.2%)	0.964
BP stage, *n* (%)				
0	8 (8.6%)	4 (9.5%)	4 (7.8%)	0.774
1	44 (47.3%)	12 (28.6%)	32 (62.7%)	0.001
2	24 (25.8%)	17 (40.5%)	7 (13.7%)	0.004
3	17 (18.3%)	9 (21.4%)	8 (15.7%)	0.478
HF class, *n* (%)				
0	5 (5.4%)	2 (4.8%)	3 (5.9%)	0.834
1	11 (11.8%)	3 (7.1%)	8 (15.7%)	0.221
2	43 (46.2%)	18 (42.9%)	25 (49.0%)	0.555
3	28 (30.1%)	16 (38.1%)	12 (23.5%)	0.130
4	6 (6.5%)	3 (7.1%)	3 (5.9%)	0.807
DM, *n* (%)	7 (7.5%)	3 (7.1%)	4 (7.8%)	0.990
Inbalanced glycemia, *n* (%)	40 (43.0%)	18 (42.9%)	22 (43.1%)	0.898
Dyslipidemia, *n* (%)	88 (94.6%)	41 (97.6%)	47 (92.2%)	0.226
CAD, *n* (%)	88 (94.6%)	40 (95.2%)	48 (94.1%)	0.811
MI, *n* (%)	2 (2.2%)	2 (4.8%)	0 (0.0%)	0.117
Pulmonary, *n* (%)	22 (23.7%)	10 (23.8%)	12 (23.5%)	0.975
Stroke, *n* (%)	5 (5.4%)	3 (7.1%)	2 (3.9%)	0.494
CKD, *n* (%)	13 (14.0%)	10 (23.8%)	3 (5.9%)	0.012
COVID-19, *n* (%)	58 (62.4%)	40 (95.2%)	18 (35.3%)	0.001

^(t)^ *t*-test for Equality of Means. Abbreviations: HTN = Hypertension; BP = Blood Pressure; HF = Heart Failure: DM = Diabetes mellitus; CAD = Coronary artery disease; MI = Myocardial Infarction; CKD = Chronic Kidney Disease.

**Table 2 life-14-01350-t002:** Clinical characteristics in relation with COVID-19 vaccination.

Characteristics	Total Cohort (*n* = 93)	Vaccination (*n* = 42)	Non-Vaccination (*n* = 51)	*p* Value for Chi-Square Test
**Signs and symptoms**				
Cough, *n* (%)	22 (23.7%)	13 (31.0%)	9 (17.6%)	0.135
Dyspnea, *n* (%)	23 (24.7%)	14 (33.3.0%)	9 (17.6%)	0.083
**Complications**				
Shock, *n* (%)	2 (2.2%)	1 (2.4%)	1 (2.0%)	0.890
Cardiac arrest, *n* (%)	2 (2.2%)	1 (2.4%)	1 (2.0%)	0.890
Days hospitalization, average ± SD (median/limits)	6.70 ± 3.07 (6/2–23)	7.12 ± 2.57 (7/3–13)	6.35 ± 3.42 (6/2–23)	0.010 ^(t)^
HAM-D, average ± SD (median/limits)	13.88 ± 3.34 (12/10–19)	13.33 ± 3.06 (13/10–19)	14.33 ± 3.52 (14/10–19)	0.152 ^(t)^
HAM-A, average ± SD (median/limits)	21.62 ± 3.37 (20/17–26)	21.19 ± 3.36 (21/17–26)	21.98 ± 3.36 (22/17–26)	0.262 ^(t)^
**Procedures performed**				
CABG, *n* (%)	4 (4.3%)	1 (2.4%)	3 (5.9%)	0.394
SYNTAX score, average ± SD (median/limits)	27.13 ± 2.78 (26.9/21.3–33.0)	27.58 ± 2.87 (27.6/21.3–32.0)	26.76 ± 2.68 (26.6/21.6–33.0)	0.153
Valvulopathy, *n* (%)	1 (1.1%)	0 (0.0%)	1 (2.0%)	0.271

^(t)^ *t*-test for Equality of Means, CABG = Coronary Artery Bypass grafting, HAM-D = Hamilton depression rating scale, HAM-A = Hamilton anxiety rating scale.

**Table 3 life-14-01350-t003:** Cardiac parameters in relation with COVID-19 vaccination.

Parameters	Vaccination Group (*n* = 42)	Non-Vaccination Group (*n* = 51)	*p* Value for *t*-Student Test
LVD d (mm)	51.79 ± 5.06	47.24 ± 12.87	0.034
LVS d (mm)	31.33 ± 7.89	30.49 ± 7.95	0.611
IVS d (mm)	12.21 ± 1.86	12.53 ± 1.96	0.432
PW d (mm)	10.52 ± 1.77	10.96 ± 2.92	0.398
EF (mm)	42.07 ± 10.46	40.39 ± 11.60	0.470
FS (mm)	19.76 ± 5.26	21.31 ± 5.94	0.190

Abbreviations and measuring units: LVD d—left ventricular diastolic diameter (mm); LVS d—left ventricular systolic diameter (mm); IVS d—interventricular septal diameter (mm); PW d—posterior wall diameter (mm); EF—ejection fraction (%); FS—fractional shortening (%).

**Table 4 life-14-01350-t004:** Percutaneous Coronary Intervention (PTCA) after COVID-19.

Characteristics	Total Cohort (*n* = 93)	Vaccination (*n* = 42)	Non-Vaccination (*n* = 51)	*p* Value for Chi-Square Test
PTCA, *n* (%)	85 (91.4%)	38 (90.5%)	47 (92.2%)	0.774
ECG STEMI, *n* (%)	87 (93.5%)	38 (90.5%)	49 (96.1%)	0.251
Non-STEMI, *n* (%)	6 (6.5%)	4 (9.5%)	2 (3.9%)	0.273
Xray Opacities, *n* (%)	28 (30.1%)	13 (31.0%)	15 (29.4%)	0.872
**Artery**				
RCA, *n* (%)	38 (40.9%)	18 (42.9%)	20 (39.2%)	0.724
LCX, *n* (%)	30 (32.3%)	16 (38.1%)	14 (27.5%)	0.277
LAD, *n* (%)	23 (24.7%)	7 (16.7%)	16 (31.4%)	0.104
LM, *n* (%)	1 (1.1%)	0 (0.0%)	1 (2.0%)	0.364
No of DES, *n* (%)				
1	57 (61.3%)	20 (47.6%)	37 (72.5%)	0.015
2	18 (19.4%)	14 (33.3%)	4 (7.8%)	0.002
3	8 (8.6%)	3 (7.1%)	5 (9.8%)	0.651
4	2 (2.2%)	1 (2.4%)	1 (2.0%)	0.890

Abbreviations: PTCA—percutaneous transluminal coronary angioplasty; ECG STEMI—ST elevation myocardial infarction electrocardiogram; Non-STEMI—non-ST elevation myocardial infarction; RCA—right coronary artery; LCX—left circumflex artery; LAD—left anterior descending artery; LM—left main artery; DES—drug eluting stent.

**Table 5 life-14-01350-t005:** Biological parameters in relation with COVID-19 vaccination.

Parameters	Vaccination Group (*n* = 42)	Non-Vaccination Group (*n* = 51)	*p* Value for *t*-Student Test
INR	1.10 ± 0.14	1.11 ± 0.19	0.740
GPT	41.48 ± 27.45	54.39 ± 40.88	0.084
GOT	93.79 ± 91.28	112.31 ± 117.40	0.406
GGT	44.55 ± 25.40	69.69 ± 14.50	0.128
Glycemia	159.76 ± 64.82	153.76 ± 68.13	0.667
CRP	27.02 ± 6.99	31.73 ± 5.79	0.602
Fibrinogen	493.05 ± 149.54	499.43 ± 124.53	0.823
Cholesterol	199.31 ± 49.04	192.14 ± 74.58	0.595
HDLc	39.90 ± 12.77	39.14 ± 13.31	0.779
LDLc	130.10 ± 44.13	124.31 ± 57.63	0.595
Tg	167.52 ± 99.60	160.63 ± 99.31	0.740
Na	189.60 ± 36.93	136.75 ± 2.49	0.048
K	4.14 ± 0.53	4.33 ± 0.38	0.047
Ureea	42.31 ± 21.66	43.43 ± 19.09	0.791
Creatinine	0.97 ± 0.19	0.98 ± 0.26	0.719
GA	11.02 ± 3.18	11.60 ± 4.19	0.462
GR	4.78 ± 0.63	4.67 ± 0.52	0.365
Hb	14.29 ± 2.03	14.01 ± 1.77	0.483
HT	41.86 ± 5.80	41.08 ± 5.08	0.490
VEM	120.42 ± 23.28	87.83 ± 5.27	0.046
HEM	29.58 ± 2.82	30.05 ± 2.04	0.345
CHEM	34.41 ± 1.71	34.34 ± 1.32	0.825
Tr	242.29 ± 62.54	234.27 ± 60.31	0.532
Galactin-3	0.73 ± 0.20 (0.73/0.42–1.18)	0.99 ± 0.18 (0.99/0.35–8.54)	0.202
Cu/ZnSOD	0.84 ± 1.02 (0.83/0.19–6.57)	0.91 ± 1.13 (0.92/0.15–6.09)	0.740
Galactin-3 (xFD)	1.46 ± 0.40 (1.45/0.84–2.37)	1.98 ± 2.59 (1.98/0.70–17.03)	0.202
Cu/ZnSOD (xFD)	167.12 ± 204.61 (167/38.64–1313)	182.12 ± 225.37 (182/30.68–1218)	0.740

## Data Availability

The original contributions presented in the study are included in the article, further inquiries can be directed to the corresponding author.

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
