# Peer review of "The Impact of COVID-19 Vaccination on Oxidative Stress and Cardiac Fibrosis Biomarkers in Patients with Acute Myocardial Infarction (STEMI), a Single-Center Experience Analysis"

_life, 2024, doi:10.3390/life14111350_

Round 1

Reviewer 1 Report

Comments and Suggestions for Authors

1.Figure 1 needs to be corrected, "no boost" means no boost, it should be the number of boosts.

2.In summary, the p-values ​​for galectin and SOD are not present.

3.p-values ​​for galectin and SOD are above 0.05, I could not find any significant change.

4.On the aspect of significancy, the statements and the results do not match.

Author Response

The authors would like to thank the area editor and the reviewers for their precious time and invaluable comments. We have carefully addressed all the comments. The corresponding changes and refinements made in the revised paper are summarized in our response below.

1.Figure 1 needs to be corrected, "no boost" means no boost, it should be the number of boosts.

R:  We thank the reviewer for this observation; the problem has been fixed.

2.  In summary, the p-values ​​for galectin and SOD are not present.

R:  We added the p-values for Galectin and SOD in summary- line 36-39.

3.p-values ​​for galectin and SOD are above 0.05, I could not find any significant change.

R:  The average level of Galectin-3 was slightly more reduced in the vaccinated patients compared to the unvaccinated patients (0.73 vs 0.99; p=0,202) and also, the average level of Cu/ZnSOD was slightly more reduced in vaccinated patients compared to the unvaccinated patients (0.84 vs. 0.91; p=0,740)- line 273-276 and line 390-397.

In our study we can confirm that oxidative stress and myocardial fibrosis biomarkers were reduced in vaccinated patients vs unvaccinated patients but with no statistical significance.

  1. On the aspect of significancy, the statements and the results do not match.

R:  The highest number of cardiovascular (CV) adverse events in patients receiving the BNT162b2 (Pfizer−BioNTech) COVID‐19 vaccine was shown in a survey based on the World Health Organization (WHO) database (VigiBase). 30% and 44% of CV adverse events were described as severe following vaccination by BNT162b2 (Pfiz-er−BioNTech) and mRNA‐1273 (Moderna). Both vaccines caused palpitations and tachycardia, as common CV adverse events [4-6]. In addition, a large group of participants without any prior cardiac disease suffered from myocarditis (n = 15) following mRNA COVID‐19 vaccination [7]. Later, a time relationship of myocarditis in young males following Pfizer−BioNTech vaccine appeared in another large‐scale study in the healthcare field [8], line 59-88.

  • In our study we mentioned all of the cardiovascular adverse events as mentioned above, also we did comparison between vaccinated vs unvaccinated patients who presented cardiovascular complication after COVID-19 infection, line 285-288.

Recent studies have shown that major complication after COVID-19 vaccination was the ischemic stroke, based on the number of outcome events during the 21‐day risk interval [16]. Among individuals with cerebrovascular disorders (n = 286) and who had recently received mRNA‐1273 (Moderna) or BNT162b2 (Pfizer−BioNTech) (n = 1,398,074), there were 246 cases of acute ischemic stroke. The fourth most common event in our study, myocarditis, has been increasingly reported in studies assessing the safety of mRNA‐1273 (Moderna) and BNT162b2 (Pfizer−BioNTech), and the majority of cases occur after the second dose [17]- line 295-302.

In the current study, heart failure within both the vaccinated and unvaccinated patients was distributed in the 4 NYHA severity degrees. We can notice that within the vaccinated group, most of them associated NYHA II heart failure (42.9%), followed by NYHA III heart failure (38.1%) and only a small part has experienced the most severe form of heart failure (7.1%). As to the group of unvaccinated patients, the hierarchy of heart failure severity hierarchy was similar: NYHA II (49%), NYHA III (23.5%), NYHA IV (5.9%), which confirms the data presented in recently published studies- line 312-318.

Thank you again for reviewing our manuscript

Reviewer 2 Report

Comments and Suggestions for Authors

Congratulations for an important work in a difficult situation with patients at high risk and, as you noticed, with questions about vaccination and Covid. I agree that vaccination did a tremendous good job but I think we need larger number of patients in order to explain the mechanism. There are a lot of variables in your study that might influenced the results.

Comments on the Quality of English Language

You have to clear some phrase: for instance line 244 “we studied an important set of parameters among which we can mention a few” - this doesn’t sound scientific and the repeated words “as regards” line 196,210, 243,262,325,332,363, or as complications. The repetitive expression is not illustrative. 

Author Response

The authors would like to thank the area editor and the reviewers for their precious time and invaluable comments. We have carefully addressed all the comments. The corresponding changes and refinements made in the revised paper are summarized in our response below.

1.  Congratulations for an important work in a difficult situation with patients at high risk and, as you noticed, with questions about vaccination and COVID. I agree that vaccination did a tremendous good job, but I think we need a larger number of patients in order to explain the mechanism. There are a lot of variables in your study that might influence the results.

R:  We thank the reviewer for this observation; we mentioned in line (409-413), study limitations can be the small number of patients included. Secondly, the small number of patients coming at cardiovascular control during Covid-19 pandemic. Study potential risks can be related to the completion of the studied group, the deterioration of blood, and the and the timely procurement of purchase kits and supplies; We are looking to continue our investigations in the near future, including a large number of both groups.

Ultimately, this syndrome cannot be considered a single clinical entity in terms of the care for patients with COVID-19 vaccination. It should be the topic for a more focused discussion and a cohesive multidisciplinary management customized to the type and severity of symptoms in correlation with comorbidities and type of vaccine- line 401-404.

2.  Comments on the Quality of English Language

You have to clear some phrase: for instance, line 244 “we studied an important set of parameters among which we can mention a few” - this doesn’t sound scientific and the repeated words “as regards” line 196,210, 243,262,325,332,363, or as complications. The repetitive expression is not illustrative.

R: We thank the reviewer for this observation; the problem has been fixed, and we also attached a certificate for translation.

Reviewer 3 Report

Comments and Suggestions for Authors

The study includes a relatively small cohort of subjects (93), with a good balance between vaccinated and not vaccinated individuals, but an inhomogeneous distribution in the number of COVID-19 comorbidity (95.2% in the vaccinated group vs 35.3% in the non-vaccinated group). The higher incidence of COVID-19 in the vaccinated group is also counterintuitive.  

There are studies that have analyzed in detail at the molecular level (metabolites and lipoprotein parameters) the effects of COVID-19 infection in the acute and post-acute phase (PLoS Pathog. 2023 Nov 9;19(11):e1011787; PLoS Pathog. 2022 Apr 21;18(4):e1010443) in vaccinated and non-vaccinated patients as well as the effect of vaccination (Front Mol Biosci. 2022 Apr 5;9:839809.).These publications should be mentioned in the Introduction and cannot be ignored when discussing data about lipid analyses.

Author Response

The authors would like to thank the area editor and the reviewers for their precious time and invaluable comments. We have carefully addressed all the comments. The corresponding changes and refinements made in the revised paper are summarized in our response below.

  1.  The study includes a relatively small cohort of subjects (93), with a good balance between vaccinated and not vaccinated individuals, but an inhomogeneous distribution in the number of COVID-19 comorbidities (95.2% in the vaccinated group vs 35.3% in the non-vaccinated group). The higher incidence of COVID-19 in the vaccined group is also counterintuitive.

R:  We thank the reviewer for this observation; we mentioned in lines 409-413 that study limitations can be the small number of patients included. Secondly, the small number of patients coming at cardiovascular control during Covid-19 pandemic. Study potential risks can be related to the completion of the studied group, the deterioration of blood, and the timely procurement of purchase kits and supplies; we are looking to continue our investigations in the near future, including a large number of both groups.

This syndrome cannot be considered a single clinical entity in terms of the care for patients with COVID-19 vaccination. It should be the topic for a more focused discussion and a cohesive multidisciplinary management customized to the type and severity of symptoms in correlation with comorbidities and type of vaccine (line 401-404).

Also, our data series were organized in the following two main study groups (STEMI patients with/without previous COVID-19 vaccination) and six subgroups based on the severity of post-COVID-19 functional status;

1.STEMI patients with previous COVID-19 vaccination:

  1. With mild post-COVID-19 functional status;
  2. With moderate post-COVID-19 functional status;
  3. With severe post-COVID-19 functional status.
  4. STEMI patients without previous vaccination anti COVID-19:
  5. With mild post-COVID-19 functional status;
  6. With moderate post-COVID-19 functional status;
  7. With severe post-COVID-19 functional status.

We want to mention that our explanation for the distribution in the number of COVID-19 comorbidities (95.2% in the vaccinated group vs 35.3% in the non-vaccinated group). The higher incidence of COVID-19 in the vaccinated group is because of the recommendation for vaccination in patients with cardiovascular comorbidities so they presented the higher risk for a second cardiac event after COVID-19 infection.

2.  There are studies that have analyzed in detail at the molecular level (metabolites and lipoprotein parameters) the effects of COVID-19 infection in the acute and post-acute phase (PLoS Pathog. 2023 Nov 9;19(11):e1011787; PLoS Pathog. 2022 Apr 21;18(4):e1010443) in vaccinated and non-vaccinated patients as well as the effect of vaccination (Front Mol Biosci. 2022 Apr 5;9:839809.).These publications should be mentioned in the introduction and cannot be ignored when discussing data about lipid analyses.

R:  We wish to thank the reviewer for bringing this paper to our attention; we discovered this work with a lot of pleasure. The mechanism proposed in this paper is also mentioned in our study in discussion section;

  • Some underlying mechanisms have been proposed for acute myocardial infarction (AMI) following COVID-19 vaccination, with Kaposi Syndrome (KS) the most likely explanation, the coexisting occurrence of acute coronary syndromes with allergic reactions. The disease has been found in four variants: (1) type I: coronary spasm in pa-tients with (nearly) normal coronary arteries; (2) type II: coronary thrombosis in pa-tients with primary asymptomatic CHD; (3) type III: stent-related allergic coronary events, with IIIa (stent thrombosis) and IIIb (in-stent restenosis); and (4) type IV: ana-phylaxis-mediated AMI in the patients with coronary grafts [23]. Aside from this, an-other possibility for AMI post COVID-19 vaccination could be vaccine-induced thrombotic thrombocytopenia (VITT) associated thrombosis, although VITT might not develop in ten minutes only, therefore, this potential is very uncommon [24]- line 342-365 and 385-389.

We acknowledge that the discussion of related work on analytical models was incomplete. We have now added several bibliographical references in introduction section, as follows;

  1. Hulscher N, Procter BC, Wynn C, McCullough PA. Clinical Approach to Post-acute Sequelae After COVID-19 Infection and Vaccination. Cureus. 2023 Nov 21;15(11):e49204. doi: 10.7759/cureus.49204. PMID: 38024037; PMCID: PMC10663976.
  2. Ghini V, Meoni G, Pelagatti L, Celli T, Veneziani F, Petrucci F, Vannucchi V, Bertini L, Luchinat C, Landini G, Turano P. Profiling metabolites and lipoproteins in COMETA, an Italian cohort of COVID-19 patients. PLoS Pathog. 2022 Apr 21;18(4):e1010443. doi: 10.1371/journal.ppat.1010443. PMID: 35446921; PMCID: PMC9022834.

15. Ghini V, Maggi L, Mazzoni A, Spinicci M, Zammarchi L, Bartoloni A, Annunziato F, Turano P. Serum NMR Profiling Reveals Differential Alterations in the Lipoproteome Induced by Pfizer-BioNTech Vaccine in COVID-19 Recovered Subjects and Naïve Subjects. Front Mol Biosci. 2022 Apr 5;9:839809. doi: 10.3389/fmolb.2022.839809. PMID: 35480886; PMCID: PMC9037139.

Thank you again for reviewing our manuscript.